# Crystal-by-Crystal Assembly in Two Types of Periodically Banded Aggregates of Poly(p-Dioxanone)

**DOI:** 10.3390/polym15020393

**Published:** 2023-01-11

**Authors:** Kuan-Ying Huang, Yu-Zhe Huang, Li-Ting Lee, Eamor M. Woo

**Affiliations:** 1Department of Chemical Engineering, National Cheng Kung University, No. 1, University Road, Tainan 701-01, Taiwan; 2Department of Materials Science and Engineering, Feng Chia University, Taichung 407-24, Taiwan

**Keywords:** poly(p-dioxanone), 3D dissection, crystallization, periodic assembly mechanisms

## Abstract

The exterior and interior lamellar assemblies of poly(p-dioxanone) (PPDO) crystallized at 76 °C yield the most regular ones to interpret the 3D assembly mechanisms and potential for structural coloration iridescence, which are investigated using atomic-force microscopy (AFM), and scanning electron microscopy (SEM). PPDO displays two types of ring-banded spherulites within a range of T_c_ with dual-type birefringent spherulites (positive and negative-type) only within a narrow range of T_c_s = 70–78 °C. At T_c_ > 80 °C, the inter-band spacing decreases from a maximum and the crystal assembly becomes irregularly corrupted and loses the capacity for light interference. Periodic grating assemblies are probed by in-depth 3D dissection into periodically banded crystal aggregates of poly(p-dioxanone) (PPDO) to disclose such layered gratings possessing iridescence features similar to nature’s structural coloration. This work amply demonstrates that grating assembly by orderly stacked crystal layers is feasible not only for accounting for the periodic birefringent ring bands with polarized light but also the distinct iridescence by interfering with white light.

## 1. Introduction

Nature’s assemblies include some common features of cyclic repetition such as spirals, periodic rings, fractals, etc. The cross-section of starch (a natural polymer), in microscopic views, displays periodic rings inter-segregated by amorphous and crystalline bands [1,2]. Crystals of synthetic polymers or compounds have long been known to pack into crystalline morphologies with some peculiar periodicity, which when viewed in polarized light microscopy, display alternate tint colors corresponding to repetition of ridge and valley as viewed in height profiles of atomic microscopy. Many inorganic or organic compounds and synthetic polymers, upon growth into crystal aggregates, often display similar periodic rhythms that are also commonly seen in Nature [3,4,5,6,7,8,9,10,11,12,13,14,15,16,17,18,19,20].

Poly(p-dioxanone) (PPDO) is a synthetic poly(ether-ester) from ring-opening polymerization of 1,4-dioxanone monomer. It has good and unique biocompatible and biodegradable properties, widely used as bio-medical materials. Owing to the ether-linkage in its chemical structure, it has better elasticity than other biodegradable polymers such as poly(L-lactic acid) (PLLA) or poly(glycolic acid) (PGA) [21]. Commercial applications of PPDO are mainly in textile fibers for medical sutures. Copolymerization of PPDO with other monomers is a common approach for widening its application properties [22,23,24]. Blending of PPDO with other polymers is also versatile in modification, such as PPDO/poly(ethylene succinate) (PESu) [25] and PPDO/poly(lactic-co-glycolic acid) (PLGA) [26]. Improvement methodology also includes incorporation of an amorphous polymer into PPDO such as PPDO/poly(vinyl phenol) (PVPh) [27,28]. Past work on PPDO had heavily focused on its biodegradability and biocompatibility, etc. Some relevant work on growth kinetics or crystalline morphology of PPDO was extensively reported in the literature [29,30,31,32,33,34,35]. However, past work documented in the literature had rarely analyzed the assembly mechanisms into periodic patterns in crystallized 3D bulk states. PPDO has been reported to display periodic birefringence bands, whose spherulite size and inter-band spacing increase with temperature within a certain temperature window [34]. As the temperature is increased to well above this temperature window, the bands become gradually corrupted and finally disappear. Such behavior is commonly seen in many other long-chain polymers as well as small-molecule compounds [3,4,5,6,7,8,9,10,11,12,13,14,15,16,17,18,19,20].

A previous work [35] has shown that as amorphous poly(p-vinyl phenol) (PVPh) was added into semi-crystalline poly(p-dioxanone) (PPDO) to induce a uniquely novel dendritic/ringed morphology. The crystal arrangement of a uniquely peculiar cactus-like dendritic PPDO spherulite displays periodic ring bands unlike those conventional types reported in the literature, but discrete and detached to self-assemble on each of the branches of the lobs. On each of the lobs, feather-like lamellae self-assemble by changing their orientation from radial to tangential. In growth cycles, the feather-like lamellae undergo periodic division into multiple branches whose number resembles the Fibonacci sequence. This means that the lamellae, in growth, do not retain a single-stalk dimension and twist continuously and monotonously from nucleus center to the outer rim. Instead they multiply, expand, and divide to fill the ever-expanding space, the branching fractals in sequence and the periodic ring-banded assembly on each of the divided lobs. The overall pattern takes the geometric shape of cactus-like dendrites, which constitute the key characteristics leading to the formation of uniquely dendritic-and-ringed PPDO spherulites. Periodically banded morphology in polymers has been widely probed in the literature. A few notable examples can be cited here: poly(1,4-butylene adipate) (PBA), poly(ε-caprolactone) (PCL), and poly([β-hydroxybutyrate] (PHB), poly(nonamethylene terephthalate) (PNT), high-density polyethylene (PE), etc. [36,37,38,39,40,41,42]. This is a continuous extension of one of our previous works on PPDO [35]. It is critical to appreciate that the periodic morphology is not only on top surfaces, but it also distributes and extends in interior assembly of 3D bulks of banded spherulites. This work was thus dedicated to using novel approaches of 3D dissection to reveal the PPDO’s periodic assemblies, whose orderly layered lamellae-stacking might be associated with light interference.

## 2. Experimental

### 2.1. Materials and Preparation

Poly(p-dioxanone) (PPDO), a biodegradable ether-ester polymer, was purchased from Sigma-Aldrich, Inc. (St. Louis, MO, USA), with viscosity = 1.5–2.2 dL/g [measured by dissolving in hexafluoroisopropanol in 0.1 % (*w*/*v*)], glass transition temperature (T_g_) = −9.5 °C, and melting temperature (T_m_) = 104.8 °C. The supplier did not provide viscosity-average molecular weight of PPDO, but it can be estimated from relevant known parameters. Müller et al. [43] and Bai et al. [44] suggested that the parameters of K and a for the Mark-Houwink equation can be used as follows: [η] = KM^a^_v_, where a = 0.63 and K = 79 × 10^3^ cm^3^ g^−1^. As only the data of viscosity = 1.5–2.2 dL/g are provided by the supplier, one can estimate the viscosity-average molecular weights (M_v_) of PPDO in this work by using the procedure provided by Müller et al. [43] and Bai et al. [44]. The estimated M_v_ of PPDO is 5.6–9.0 × 10^4^ g/mol.

PPDO was dissolved into p-dioxane (a good solvent for PPDO) to 4 wt.% polymer solution, stirred till homogeneous; then the homogeneous solution was cast on the glass slide as thin film, and dried by placing at 30 °C in a vacuum oven to remove the residual solvent. Film samples were first heated to a maximum melting temperature (T_max_) for 2 min for erasing the prior thermal history. The specimens at molten state were quickly transferred from the hotplate to a precise temperature-controlled microscopic hot stage (temperature precision ± 0.5 °C), which was pre-equilibrated at specified crystallization temperatures till completion.

### 2.2. Apparatus

Polarized-light optical microscopy (POM, Nikon Optiphot-2, Tokyo, Japan), with a Nikon Digital Sight (DS-U1) camera and a microscopic hot stage (Linkam THMS-600 with T95 temperature programmer, Linkam Scientific Instruments Ltd., Surrey, UK), was used. A wavelength-tint plate (λ = 530 nm) was inserted in POM to make contrast interference colors for all POM graphs.

Atomic-force microscopy (AFM). Intermittent tapping mode of AFM (diCaliber, Veeco Co., Santa Barbara, CA, USA) with a silicon-tip (f = 70 kHz, r = 10 nm) was used. The scanner range was 150 × 150 μm, and for larger magnifications on selected areas of interest, the scan range could be zoomed in to 5 × 5 μm.

For the high-resolution field-emission scanning electron microscopy (Hitachi SU8010, HR-FESEM), samples were coated with platinum using vacuum sputtering (10 mA, 300 s) prior to SEM observation. For analysis of interior lamellar assembly using SEM, the bulk-crystallized specimens (on the glass slide) were fractured across the film thickness. Fracturing was conducted by precut using a diamond knife across the back side of the specimens that were pre-coated on glass slides. When necessary, pre-quenching into a liquid nitrogen bath was performed to ensure that there was no ductile straining prior to breaking the films. Top surfaces and interior of crystallized samples were analyzed for full 3D architectures. Lamellar architectures of interiors were fully exposed by such dissection techniques, which were unique and critically novel in comparison to conventional morphology investigations on the top surfaces of film specimens only.

## 3. Results and Discussion

PPDO samples were heated to T_max_ = 150 °C for 2 min (for erasing prior thermal histories), then rapidly quenched to specific T_c_ for isothermal crystallization till full crystallinity. Figure 1 shows POM micrographs of birefringence patterns at various temperatures of crystallization (T_c_’s). As T_c_ is increased from 60 to 70 °C (Figure 1a–c), the inter-band spacing increases to a maximum. At T_c_ < 70 °C, the inter-band spacing is narrow between 4–5 μm, and the spherulites belong to “negative-type” birefringence. At T_c_ = 70–78 °C, the inter-band spacing rapidly expands, and peculiarly, two birefringence types co-exist (Figure 1d–f). For convenience, these two spherulitic optical types are termed as Type-p and Type-n (corresponding to positive- and negative-type, respectively). Results of POM morphology for PPDO crystallized at higher temperatures (>80 °C) are placed in Appendix A. At T_c_ > 80 °C, the inter-band spacing rapidly increases and the band regularity becomes corrupted. At T_c_ = 90 °C, the bands become almost corrupted/disintegrated, but the successive rings are still discernible and the inter-band spacing decreases from the previous maximum.

Figure 1c,e,f also shows that the POM images of ring-banded spherulites of PPDO crystallized at T_c_ = 70, 74, 76, and 78 °C, respectively, can display two different and opposing birefringence types simultaneously co-existing: positive-type (Type-p) and negative type (Type-n). For the positive type, the first and third quarters of the spherulites display alternate blue-orange stripes, with crystals of blue birefringence stripes dominating the rings. For the negative type (Type-n), the first and third quarters of the PPDO spherulites display reversed alternate blue-orange stripes, with crystals of orange birefringence stripes dominating the rings. POM images only yield general birefringence patterns; thus, for detailed morphology, further SEM and AFM analyses are to be discussed in the following sections. It may be puzzling as to why two types of banded spherulites co-exist at a same T_c_, which requires some rationale. Packing and maturing into multiple (or dual) types of spherulites may be related to nucleation instability, where the initial geometry of nucleus/nuclei (or bundled sheaves) serves as a determining factor for guiding the later lamellae to grow and assemble. In several other polymers, it has been also found that dual or multiple types of spherulites co-exist due to the instability of the initial nucleus, as documented in several literature papers [45,46].

Both positive-type and negative-type birefringence PPDO spherulites co-exist when crystallized at a specific T_c_ from 70 to 78 °C, but the relative fraction may vary slightly. Figure 2 shows the relative percentage change of negative-type vs. positive-type ring-banded PPDO spherulites with respect to T_c_. Numerical values of the relative percentages of these two types are listed in Appendix A. The percentage of Type-p is dominant over that of Type-n; Type-p ranges from 65% to 75% as PPDO is crystallized between T_c_ = 70–78 °C. A peculiar exception occurs when PPDO is crystallized at T_c_ = 76 °C, where Type-p spherulites are the majority and reach 96% (estimated average percentage).

Inter-band spacing usually varies with T_c_ as reported in many polymers displaying periodic bands. Peculiarly for PPDO spherulites, the inter-band spacing does not vary monotonically with T_c_; it steadily increases then goes through a maximum at T_c_ = 80 °C, with a sharp cusp, and finally decreases with further T_c_ increases. A plot of inter-band spacing vs. T_c_ is shown in Figure 3. The maximum inter-band spacing occurs at T_c_ = 80 °C, where the ring pattern becomes highly corrupted. The most regular ring bands occur at T_c_ = 75–78 °C. A dramatic increase in inter-band spacing occurs in a temperature window of T_c_ = 78 to T_c_ = 80 °C; after the maximum, a sharp decrease in the spacing occurs at T_c_ = 82–84 °C. Note that average values of inter-band spacing for Type-p and Type-n are used in the plot.

In situ monitoring of growth of ring-banded PPDO spherulites are shown in Appendix A, which displays in situ POM graphs captured at T_c_ = 76 °C at different times of crystallization by quenching from T_max_ = 150 °C. At initiation, the nuclei are initially spherical shaped, where emerging lamellae radiate outward from a common center to pack into periodically banded aggregates. The interior assembly will be discussed in details later using SEM dissection analysis. By ignoring the later-stage difference between Type-n and Type-p, focus is directed to the interior morphology in the valley and ridge of PPDO ring-banded spherulites crystallized at T_c_ = 70–76 °C, where the most regular ring bands occur. Figure 4 shows (a) POM graph, (b) AFM height image zoom-in to yellow square in Graph-(a), and (c) height profile of PPDO crystallized at T_c_ = 76 °C. By side-by- side comparison of the POM vs. AFM images (Figure 4a,b) on the counterpart zones, one is able to discern on the first quarter of a single spherulite in POM, the ridge bands correspond to orange-birefringence color rings, and the valley bands correspond to blue-birefringence color rings. The arrow on AFM height image marks the radial direction from its nucleus center. Along this arrow direction, AFM height profile is constructed and shown in Figure 4c. Average band spacing (BS) is determined to be 16 μm for PPDO at T_c_ = 76 °C. The average height drop from ridge and valley is ca. 400 nm, which is much smaller than the dimensional width (~2–3 um) of a typical single-crystal lamella. Thus, it is not feasible to regard the periodic height drop as associated with the lamellae undergoing a continuous screw-like helix. Interior dissection to expound more truthful assembly mechanisms is to be presented in latter sections of SEM characterization.

The PPDO spherulites at higher T_c_ possess increasingly wider inter-band spacing but lesser regularity. AFM images could yield clear height profile and surface textures of lamellar assembly. PPDO specimens crystallized at T_c_ = 76 °C were selected for deeper analysis as they gave wide enough inter-band spacing but still retained reasonable regularity. Figure 5 shows zoom-in AFM (a) (c) height and (b) (d) phase images at greater magnification of the periodic ring bands of PPDO (Type-p) crystallized at T_c_ = 76 °C. AFM images reveal clearly fine-fibrous textures of lamellae on the ridge, but the textures on the valley are not so obvious even in phase images. Nevertheless, one can tell that there are no visible continuously helix-twist lamellae from ridge to valley zones, as the lamellae all retain a similar fibrous character and there is only up-and-down height variation in the radial direction.

AMF microscopy results alone may not be sufficient. AFM analysis is further coupled and compared with the result of SEM analysis on the top surfaces of PPDO at T_c_ = 76 °C, which is shown in Figure 6a,b for side-by-side comparison of AFM and SEM morphologies, respectively. The inter-band spacing is consistent between the AFM and SEM result, being ~12 μm, which in turn agrees with that estimated from the POM images shown earlier for PPDO at T_c_ = 76 °C. The SEM image yields a fact that as the lamellae rise up from the valley to ridge, they become slightly wider (1–2 μm in width); they also bend toward the circumferential direction. Figure 6c shows a simplified scheme for the top-surface lamellar textures of banded PPDO. Once again, there is no evidence of continuously helix-twist lamellae on the top surfaces.

For comparison of Type-n vs. Type-p ring-banded PPDO at T_c_ = 70–78 °C, they were respectively analyzed using AFM. For simplicity, only Type-n PPDO at T_c_ = 70 °C was investigated in details; Type-p at T_c_ = 70 °C was similarly investigated in brief. Figure 7a–c shows AFM height/phase images, and height profiles, and (d–f) zoom-in phase images of Type-n PPDO ring-banded spherulites crystallized at T_c_ = 70 °C.

Similarly, by comparison between Type-n and Type-p ring bands, Figure 8a–c shows AFM height/phase images, and height profiles, respectively, and Figure 8d–f shows zoom-in AFM phase images of Type-p PPDO ring-banded spherulites crystallized at T_c_ = 70 °C. Although the depth profiles are similar between Type-n and Type-p PPDO spherulites, the detailed lamellar assembly appears to be different in these two types. The lamellae in Type-p PPDO ring-banded spherulites apparently display a sharp cusp between successive bands in the transition zone, where the lamellae are dramatically oriented in the upward tangential direction. By contrast, such feature is absent in Type-n PPDO ring-banded spherulites. For brevity, AFM height-profile analyses on possible differences between PPDO Type-n and Type-p ring-banded spherulites at all three different T_c_’s (70, 78, and 85 °C) are compared and summarized in Appendix A.

To further confirm the earlier AFM analysis on top surfaces of Type-n vs. Type-p, SEM characterization was performed on the fractured interiors of these two types of PPDO ring-banded spherulites. Figure 9 shows SEM images zoomed-in to interior crystal assemblies for (a,b) Type-n and (c,d) Type-p PPDO ring-banded spherulites crystallized at T_c_ = 70 °C. The general features or interior morphology appear to be similar in that both show periodic lamellar orientation changes with respect to the alternate valley and ridge zones. The interior lamellae in the Type-n PPDO banded spherulites display an oblique angle as the lamellae traverse from ridge to valley zone; by contrast, the interior lamellae in the Type-p PPDO banded spherulites tend to have a normal (mutually at ~90° angle) intersection.

It is critical to probe into the transitional zone from ridge to valley, and examine carefully how lamellae traverse from ridge to valley in interior assembly. Figure 10a shows an SEM micrograph of PPDO spherulite (T_c_ = 76 °C) fractured tangentially covering a transitional zone from the ridge to valley band along the tangential direction, where the interior lamellae have gradient oblique inclinations as they traverse from ridge to valley bands. Underneath the top-surface ridge zone, obviously, all lamellae are oriented normally (with respect to substrate). The fracture interior demonstrates that interior lamellae undergo an oblique angle tilting and an inverted “U-bend” as they move from ridge to valley. The inverted “U-bend” constitutes a flat plateau on the top surface of the valley band, and the normal orientation of the lamellae constitutes the ridge band. Figure 10b shows POM micrograph of the corresponding optical pattern of periodic bands, and Figure 10c is a scheme illustrating the periodic up-and-down on top surface and the interior lamellae going through normal-tangential orientation alteration and periodic bend underneath the top-surface ridge-valley zones.

It would be interesting and essential to justify the universal assembly by comparing the interior-dissection of neat PPDO of this work to that of PPDO blended with a miscible diluent, poly(ethylene succinate) (PESu), as demonstrated in an earlier sequential work [48]. As the neat PPDO is blended with amorphous and miscible PESu, the self-assembled interior lamellae become more clearly distinct and it is more feasible to probe into the architectures with respect to the transitional zone from ridge to valley, and examine carefully how the lamellae traverse from ridge to valley in the interior assembly. Figure 11 shows SEM graphs for interior 3D periodic assemblies of PPDO/PESu (75/25) crystallized at T_c_ = 75 °C [49]. The SEM micrograph more vividly reveals the dissected interior revealing horizontal-oriented lamellae are assembly underneath the top-surface valley, and normal-oriented ones are underneath the top-surface ridge zone, with a transition zone where lamellae are at oblique angles as they traverse from ridge to valley. Alternate normal to horizontal lamellae are respectively located underneath the top-surface ridge and valley zones. Once again, the interior 3D dissection uniquely testifies that the periodic height rise-and-drop and accompanying rings are not associated with the interior lamellae undergoing a continuous screw-like helix twist.

To summarize, Figure 12 shows two schematics illustrating a 3D correlation of top surface versus fractured interior lamellae in the periodically banded PPDO, which are exposed in two fracture directions: (a) to cut across both radial and tangential directions, and (b) to cut along the radial direction only, respectively. Grating assembly is obvious, with each cross-bar pitch being clearly composed of alternate layers of crystal plates with perpendicular-horizontal intersection. Figure 12a shows that fracture occurs along a slant angle across both radial and tangential directions where a portion of the radial-fracture view is accompanied with a tangential-fracture view. Fracture lines across specimens might be incidental and case-by-case; in this case, the tangential fracture happened to cut along a circumferential ridge band, where normal-oriented lamellae are positioned directly underneath each of the top-surface ridge bands. By contrast, interior dissection across the radial direction as exemplified in scheme of Figure 12b clearly shows that the radial-oriented lamellar plates, with an inverted “U” curvature, are in a horizontal position (with respect to the substrate glass) and are located directly underneath each of the valley bands.

It has been proven that grating-assembled lamellae in ring-banded crystals of some other polymers have the potential to act as light-interference media to produce iridescence [38,39] resembling the color spectrum from nacre (pearls or abalone shells) or opal. The specimens of neat PPDO crystallized as both Type-p and Type-n periodic bands were then tested for iridescence using experimental procedures and set-ups similar to those used in the cited work [38,39]. Top-row graphs in Figure 13 shows iridescence features of neat PPDO banded spherulites (of both Type-p and Type-n) are evident when crystallized in range of T_c_ (60, 65, 67 °C) where orderly ring bands are present (portions of POM images shown for references as insets). By interfering with white light, the orderly layered-lamellae in the banded PPDO aggregates all display striking pearl-like shiny coloration. Note that the inter-band spacing ranges from 4.0–4.8 µm in these banded PPDO, whose micro-structures and height profiles are suited for light interference according to analyses in earlier work [38,39]. This periodicity in orderly layers with tuned nano-/micro-dimensions reveals that the interior and top-surface lamellar crystals are able to cause white light to be interfered with the periodically grating-assembled aggregates as the light reflects or passes through the crystals’ orderly arrays.

## 4. Conclusions

In a specific crystallization temperature range, PPDO exhibits two co-existing types of ring-banded spherulites, which were characterized to reveal the crystalline arrangement on the top surface. The crystalline arrangements of these dual types of ring-banded spherulites at T_c_ = 76 °C are similar. The ridge region is composed of broad branching crystals that grow but taper to form thin branches in the valley region. In the interior of the PPDO banded spherulites, the plate-like lamellae bend and turn to perpendicular to the substrate and form the dense bundles periodically. The periodic lamellar assembly leads to the ring-banded pattern composed of branching on the exterior top surface. The PPDO crystalline morphology demonstrates that periodically fractal branching and discontinuous interior assembly are the main characteristics leading to formation of the ring-banded spherulites. In addition, the work proves that the orderly grating-assembled lamellae in periodic arrays of crystalline PPDO display colorful iridescence just like that seen in mineral-gem opals that are crystals of 150–300 nm microspheres packed in periodic ordering. Furthermore, the conventionally thought models of continuous helix-twist lamellae may not be an exclusive proposition for interpreting the periodicity; this work demonstrates that grating assembly by orderly-stacked crystal layers is more feasible for accounting for the birefringent ring bands with polarized light as well as distinct iridescence by interfering with white light.

## Figures and Tables

**Figure 1 polymers-15-00393-f001:**
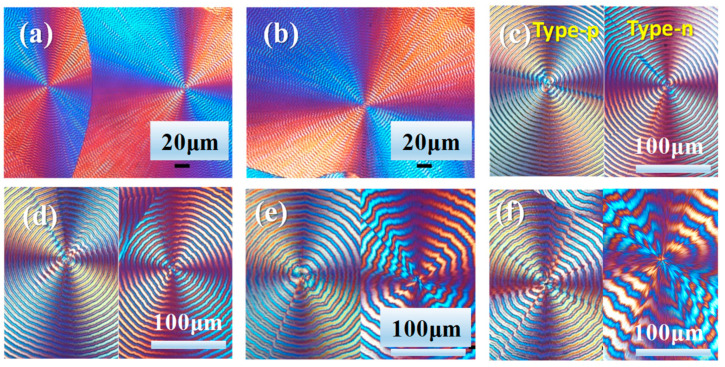
POM graphs of PPDO crystallized at various T_c_: (**a**) 60 °C, (**b**) 65 °C, (**c**) 70 °C, (**d**) 74 °C, (**e**) 76 °C, (**f**) 78 °C.

**Figure 2 polymers-15-00393-f002:**
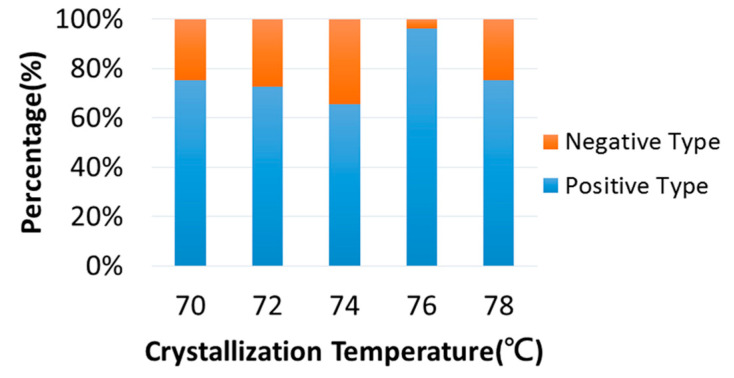
Bar charts for relative percentage change with respect to T_c_ for Type-n vs. Type-p ring-banded PPDO spherulites.

**Figure 3 polymers-15-00393-f003:**
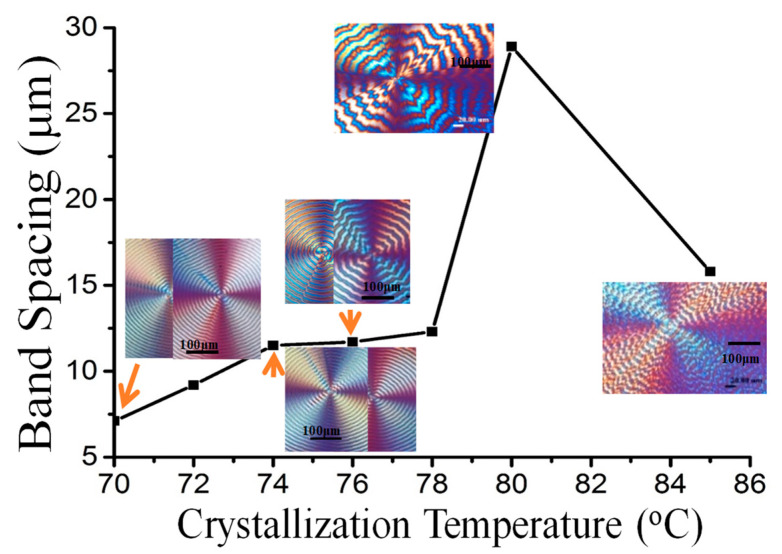
Inter-band spacing variation of banded PPDO with respect to T_c_. (average value of inter-band spacing is taken for Type-p and Type-n).

**Figure 4 polymers-15-00393-f004:**
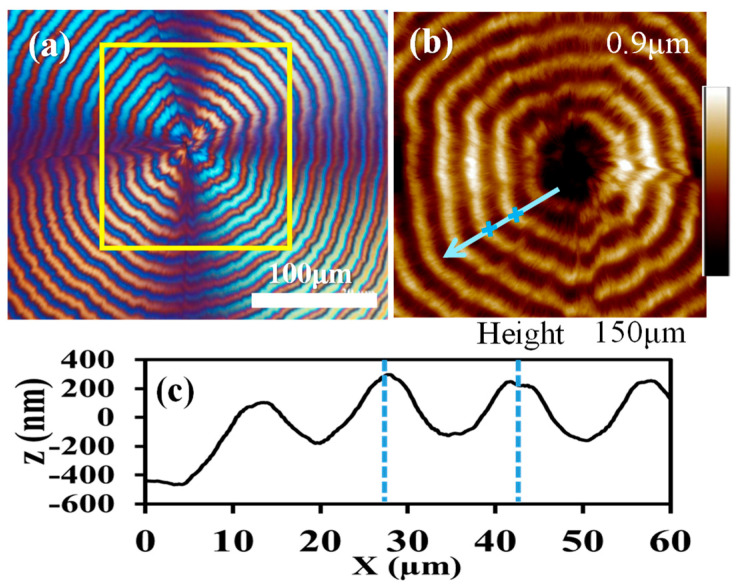
(**a**) POM graph, (**b**) AFM height image zoom-in to yellow square in Graph-(**a**,**c**) height profile of PPDO crystallized at T_c_ = 76 °C.

**Figure 5 polymers-15-00393-f005:**
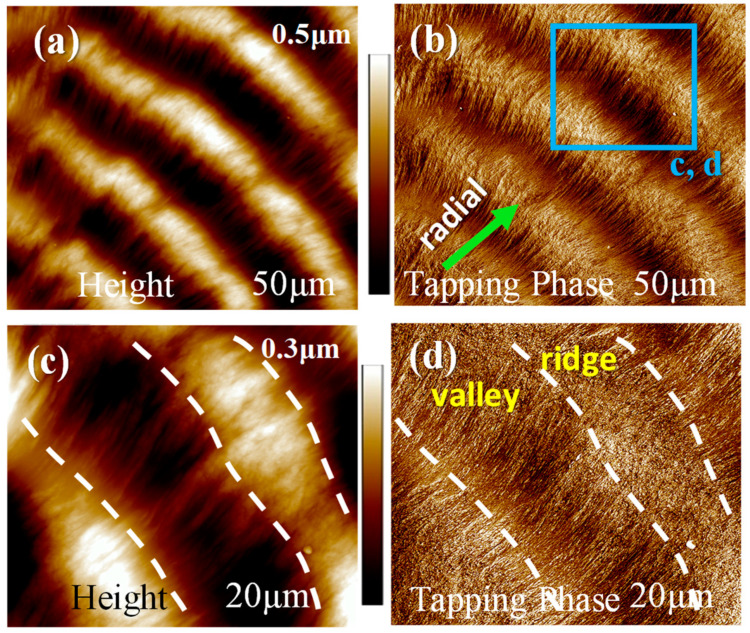
AFM (**a**,**c**) height and (**b**,**d**) phase images of the periodic ring band of PPDO crystallized at T_c_ = 76 °C.

**Figure 6 polymers-15-00393-f006:**
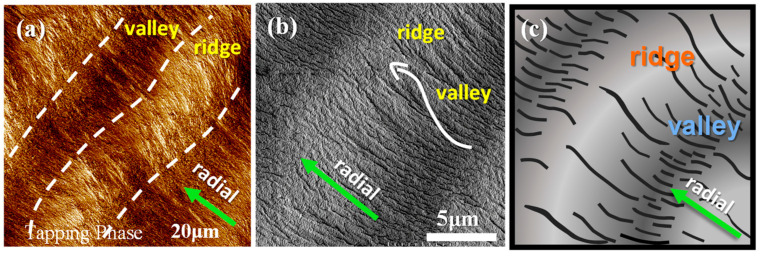
(**a**) AFM phase image, (**b**) SEM micrograph, and (**c**) scheme of the crystalline arrangement on the top surface of PPDO crystallized at T_c_ = 76 °C. Reprinted with permission from Ref. [47]. Copyright 2023 American Chemical Society.

**Figure 7 polymers-15-00393-f007:**
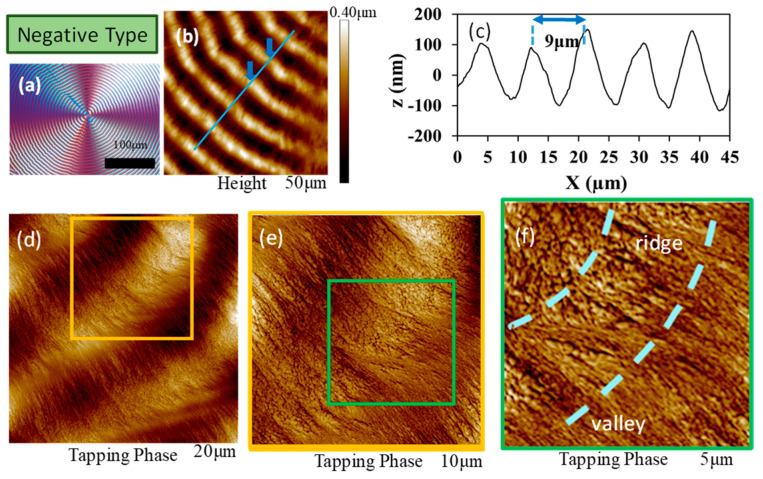
(**a**–**c**) AFM height/phase images, and height profiles, (**d**–**f**) zoom-in phase images of Type-n PPDO ring-banded spherulites crystallized at T_c_ = 70 °C.

**Figure 8 polymers-15-00393-f008:**
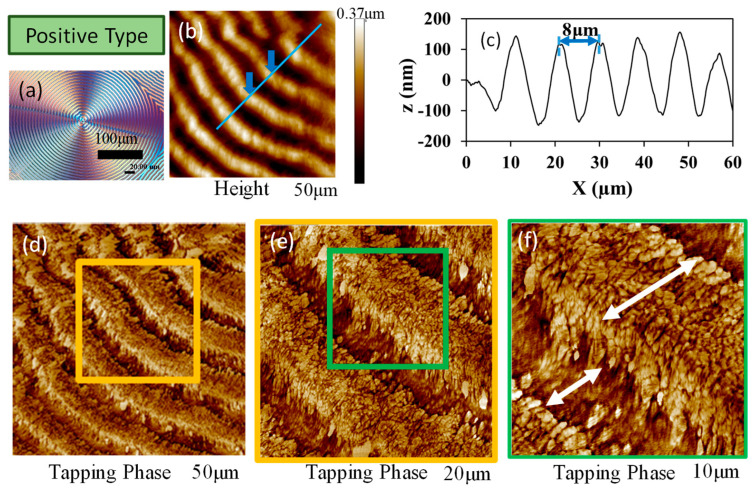
(**a**–**c**) AFM height/phase images, and height profiles, respectively, and (**d**–**f**) step-wise zoom-in phase images of Type-P PPDO ring-banded spherulites crystallized at T_c_ = 70 °C.

**Figure 9 polymers-15-00393-f009:**
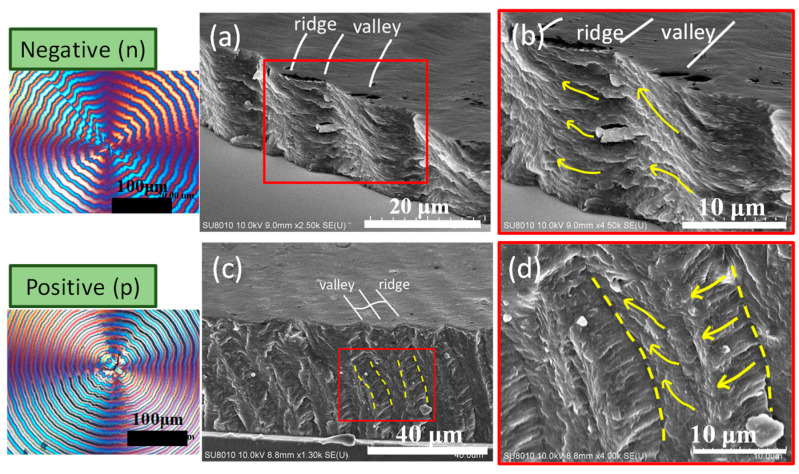
POM and SEM images zoom-in to interior crystal assemblies for (**a**,**b**) Type-n and (**c**,**d**) Type-p PPDO ring-banded spherulites crystallized at T_c_ =70 °C. Insets at left are POM images for two types of PPDO bands.

**Figure 10 polymers-15-00393-f010:**
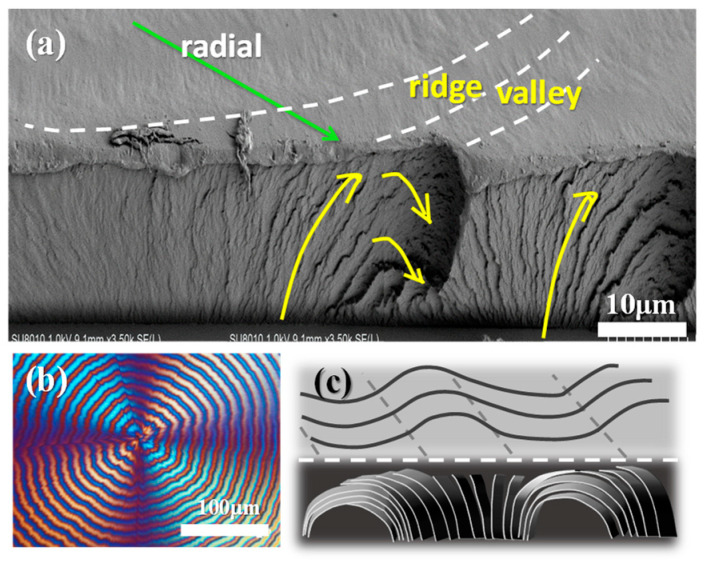
(**a**) SEM micrographs of neat PPDO spherulite (T_c_ = 76 °C) fractured tangentially covering a transitional zone from the ridge to valley band along the tangential direction, showing gradient-oblique inclinations of lamellae, (**b**) POM micrograph, and (**c**) scheme illustrating the lamellar U-bend underneath valley zone.

**Figure 11 polymers-15-00393-f011:**
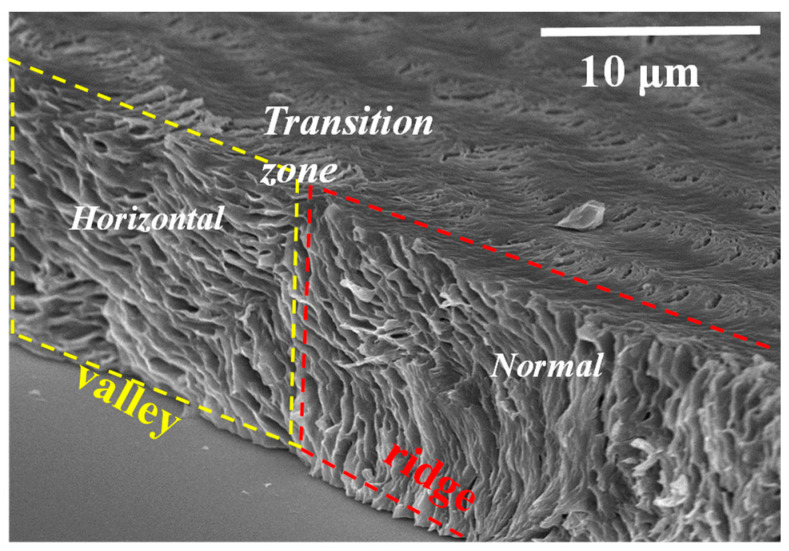
SEM graphs for interior 3D periodic assemblies, revealing alternate normal-horizontal lamellae underneath the top-surface ridge and valley zones of PPDO/PESu crystallized at T_c_ = 75 °C [46].

**Figure 12 polymers-15-00393-f012:**
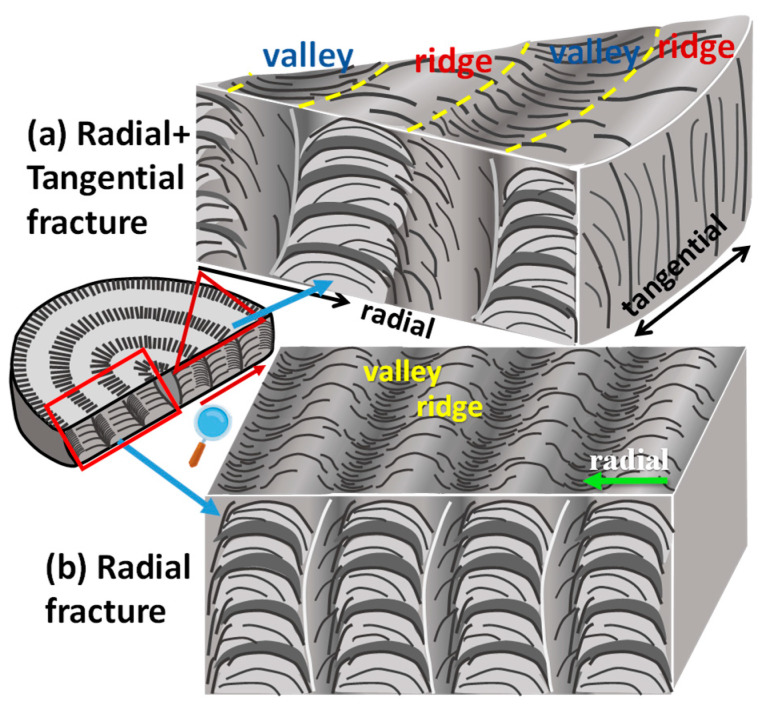
Schematics for PPDO crystal aggregates’ top-surface periodic banding in correlation with interior lamellar assembly fractured along (**a**) both radial and tangential directions vs. (**b**) radial direction only.

**Figure 13 polymers-15-00393-f013:**
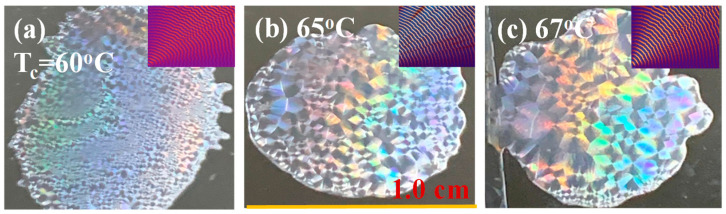
Bright pearl-like iridescence of neat PPDO banded spherulites of both type-p and -n cast on glass substrates (1.0 cm × 1.0 cm); representative POM images (insets on upper-right) with orderly ring-bands at T_c_ = (**a**) 60 °C, (**b**) 65 °C, and (**c**) 67 °C, respectively.

## Data Availability

Data are contained within the article and are available upon reasonable request.

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
