# Peer review of "Crystal-by-Crystal Assembly in Two Types of Periodically Banded Aggregates of Poly(p-Dioxanone)"

_polymers, 2023, doi:10.3390/polym15020393_

Round 1

Reviewer 1 Report

The article "Crystal-by-Crystal Assembly in Two Types of Periodically Banded Aggregates of Poly(p-dioxanone)" describes the obtained structural pattern of PPDO that can generate structural color. It is a valuable study that can be published after authors address the following problems:

Abstract should be checked and revised carefully by briefly introducing the work plan and key findings. Abstracts should highlight the innovation of the article, as often abstract section is presented separately in search engines, it must be able to stand alone as an informative piece.

Abbreviations must be explained at first use (Tc, TcS, Tc`S, Tg, Tm). The meaning might be obvious for readers with DSC knowledge but let’s assume a broader readership. In addition, if authors have access to a DSC, measurements should be done for adding value to the paper.

The English language needs some minor polishing for style and typos.

In Figure 1 the size bars should be remade so that images a and b don’t present a former bar information remains.

At row 179 change um to mm.

What is the novelty and intended uses? Motivation must be clear and authors should explain it. The conclusion should reflect the heuristic of the study.

Author Response

Referee #1 - Comments and Suggestions for Authors

The article "Crystal-by-Crystal Assembly in Two Types of Periodically Banded Aggregates of Poly(p-dioxanone)" describes the obtained structural pattern of PPDO that can generate structural color. It is a valuable study that can be published after authors address the following problems:

  1. Abstract should be checked and revised carefully by briefly introducing the work plan and key findings. Abstracts should highlight the innovation of the article, as often abstract section is presented separately in search engines, it must be able to stand alone as an informative piece.

Authors responses/revision: Thanks for suggestion. We have re-phrased the texts to shed more clear focus of novelty and essence of key findings. Abstract has been totally re-structured, as shown in blue-font texts. Referee’s comments have been very constructive and helpful to elevate the novelty and scientific quality.

  1. Abbreviations must be explained at first use (Tc, TcS, Tc`S, Tg, Tm). The meaning might be obvious for readers with DSC knowledge but let’s assume a broader readership. In addition, if authors have access to a DSC, measurements should be done for adding value to the paper.

Authors responses/revision: Thanks for suggestion. We have searched all abbreviations and fixed them, per the kind comments.

  1. The English language needs some minor polishing for style and typos.

Authors responses/revision: Thanks for suggestion. We have more carefully polished the writing to avoid careless typos, per the kind comments.

  1. In Figure 1 the size bars should be remade so that images a and b don’t present a former bar information remains.

Authors responses/revision: Thanks a lot for suggestion and keen observation. Changes have been done, per the kind comments. The originals were indeed not so aesthetic. We are grateful for the opportunity for us to improve the figure images and labeling.

  1. At row 179 change um to mm.

Authors responses/revision: Thanks, we think that probably referee refers to the writing of “The dimensional width (~2-3 um) of a typical single-crystalline lamella”. However, here we meant to state the “typical dimension of a single crystal” like polymer grown from dilute solutions. Classical textbooks indicate that a single crystal (e.g., PE grown from Xylene) has width x length = 3-5 um, and thickness = 10-20 nm. Here we did not mean to state the length of spherulite’s diameter that can grow up to as large as 2-3 mm. Hope our clarification addresses Referee’s question?

  1. What is the novelty and intended uses? Motivation must be clear and authors should explain it. The conclusion should reflect the heuristic of the study.

Authors responses/revision: Thanks for suggestion. Similar to our reply to Comment-1 re. abstract, we have re-written the Conclusion texts to shed more clear focus of novelty and essence of key findings (blue-font text). Referee’s comments have been very constructive and helpful to elevate the novelty and scientific quality.

Reviewer 2 Report

This manuscript investigated the temperature-dependent ring-banded spherulites of poly(p-dioxanone) (PPDO), and their periodically banded crystal aggregates and assembly mechanisms. The reviewer suggests minor revision to this manuscript.

In Figure 1, both positive-type and negative-type birefringence spherulites coexist when PPDO crystallized at a high Tc (74-78 °C), and their relative fraction vary slightly with Tc. What causes the co-existence of positive-type and negative-type spherulites of PPDO? Also, the negative-type spherulites of PPDO have largely distinct inter-band spacing with their positive-type spherulites, especially for PPDO crystallized at 78 °C in Figure 1f. This should be explained.

Since the negative-type and positive-type spherulites of PPDO have remarkable different inter-band spacing in Figure 1, the author how to calculate the inter-band spacing of banded PPDO at different Tc in Figure 3? It is recommended to be indicated.

In Figure 3, when Tc is lower than 80 °C, the inter-band spacing of banded PPDO increases with Tc. However, in Figure S3, AFM profiles show that both Type-p and Type-n of PPDO crystallized at Tc = 76 °C have a large inter-band spacing than those of PPDO crystallized at Tc = 78 °C.

Author Response

Referee #2 - Comments and Suggestions for Authors

This manuscript investigated the temperature-dependent ring-banded spherulites of poly(p-dioxanone) (PPDO), and their periodically banded crystal aggregates and assembly mechanisms. The reviewer suggests minor revision to this manuscript.

  1. In Figure 1, both positive-type and negative-type birefringence spherulites coexist when PPDO crystallized at a high Tc(74-78 °C), and their relative fraction vary slightly with Tc. What causes the co-existence of positive-type and negative-type spherulites of PPDO? Also, the negative-type spherulites of PPDO have largely distinct inter-band spacing with their positive-type spherulites, especially for PPDO crystallized at 78 °C in Figure 1f. This should be explained.

Authors responses/revision: Thanks a lot for suggestion and keen observation. Packing and maturing into multiple (or dual) types of spherulites may be related to nucleation instability, where the initial geometry of nucleus/nuclei (or bundled sheaves) is a determining factor for lamellae to grow and assemble. In many polymers systems, it has been common that dual or multiple types of spherulites co-exist. See for examples in the following three papers:

  1. {ref. 41 as already cited} C.-H. Tu, E. M. Woo, S. Nagarajan and G. Lugito, “Sophisticated Dual-Discontinuities Periodic Bands of Poly(nonamethylene terephthalate)”, CrystEngComm., 23, 892-903 (2021). And other two:
  2. L. Tseng, K. -N. Chuan, and E. M. Woo, “Unusual Ringed/Dendritic Sector Faces in Poly(butylene succinate) Crystallized with Isomeric Polymer”, Ind. Eng. Chem. Res., 59, 7485−7494 (2020).
  3. Y. -T. Yeh and E. M. Woo, “Anatomy into Interior Lamellar Assembly in Nuclei-Dependent Diversified Morphologies of Poly(L-lactic acid)”, Macromolecules, 51 (19), 7722-7733 (2018).

We have added such proposed interpretation in R1.

Re. Referee’s comment on Fig. 1f – “the negative-type spherulites of PPDO have largely distinct inter-band spacing with their positive-type spherulites. We think that these two types vary differently with respect to temperature of crystallization. With Tc increase, Type-n ring bands simply mature and then collapse into irregular bands faster than Type-p. àWe have added texts of explanation in R1.

  1. Since the negative-type and positive-type spherulites of PPDO have remarkable different inter-band spacing in Figure 1, the author how to calculate the inter-band spacing of banded PPDO at different Tcin Figure 3? It is recommended to be indicated.

Authors responses/revision: Thanks for keen observation and suggestion. We did not distinguish the inter-band difference between these two types (maybe we should have), as our focus was on the morphology difference/contrast, and not so much on the inter-band spacing variation. We took the average of inter-band spacing of these two types and plotted the value in Fig. 3 (band spacing), which is now indicated on the figure caption.

  1. In Figure 3, when Tcis lower than 80 °C, the inter-band spacing of banded PPDO increases with Tc. However, in Figure S3, AFM profiles show that both Type-p and Type-n of PPDO crystallized at Tc = 76 °C have a large inter-band spacing than those of PPDO crystallized at Tc = 78 °C.

Authors responses/revision: Figure S3, AFM profiles: We double-checked our raw AFM data, and found that Fig. S3 was not correctly labeled in temperature. The caption should have been: Tc: (a) 70 oC, (b) 78 oC, and (c) 85 oC. We think that this should address the referee’s question. At Tc=85oC, the inter-band spacing decreases and becomes smaller than those at Tc=78oC, and the height drop (profile) also decrease from that at Tc=78oC. We have corrected the errors in R1.